# Measuring health-related quality of life of care home residents, comparison of self-report with staff proxy responses for EQ-5D-5L and HowRu: protocol for assessing proxy reliability in care home outcome testing

Adeela Usman,[1] Sarah Lewis,[2] Kathryn Hinsliff-Smith,[1] Annabelle Long,[1] Gemma Housley,[3] Jake Jordan,[4] Heather Gage,[4] Tom Dening,[5] John R F Gladman,[6,7,8] Adam L Gordon[1,7,8,9]

For numbered affiliations see end of article.

**Correspondence to**
Dr Adam L Gordon;
adam.gordon@nottingham.ac.uk

## ABSTRACT

**Introduction** Research into interventions to improve health and well-being for older people living in care homes is increasingly common. Health-related quality of life (HRQoL) is frequently used as an outcome measure, but collecting both self-reported and proxy HRQoL measures is challenging in this setting. This study will investigate the reliability of UK care home staff as proxy respondents for the EQ-5D-5L and HowRu measures.

**Methods and analysis** This is a prospective cohort study of a subpopulation of care home residents recruited to the larger Proactive Healthcare for Older People in Care Homes (PEACH) study. It will recruit residents ≥60 years across 24 care homes and not receiving short stay or respite care. The sample size is 160 participants. Resident and care home staff proxy EQ-5D-5L and HowRu responses will be collected monthly for 3 months. Weighted kappa statistics and intraclass correlation adjusted for clustering at the care home level will be used to measure agreement between resident and proxy responses. The extent to which staff variables (gender, age group, length of time caring, role, how well they know the resident, length of time working in care homes and in specialist gerontological practice) influence the level of agreement between self-reported and proxy responses will be considered using a multilevel mixed-effect regression model.

**Ethics and dissemination** The PEACH study protocol was reviewed by the UK Health Research Authority and University of Nottingham Research Ethics Committee and was determined to be a service development project. We will publish this study in a peer-reviewed journal with international readership and disseminate it through relevant national stakeholder networks and specialist societies.

## INTRODUCTION

Long-term care facilities in the UK are called care homes and are classified as either care homes with or without nursing based on the availability of registered nurses on-site.[1] The types of residents cared for in both classifications of facility are similar and all UK care homes are included in the international consensus definition of a nursing home.[2] Around 425 000 people live in care homes in the UK[3] with most residents requiring care due to disability from long-term conditions. The majority of residents are aged over 85, 75–80% of residents live with dementia[4] and over half of residents die within 12 months of admission to care home.[5]

### Strengths and limitations of this study

► To our knowledge, this is the first study to investigate the reliability of staff proxy using the EQ-5D-5L and HowRu and the reliability of HowRu as a quality of life measure in a UK care home setting.
► The scales will be administered to residents and staff on the same day, thus reducing any variations that will affect the level of agreement.
► Analysing the characteristics of staff respondents will help to generate insights into how proxies might better be selected or prepared to maximise the validity of proxy responses.
► The use of multiple time points will help to increase the sample size and account for the influence of staff turnover on the quality of proxy response.
► Working with residents with mental capacity will optimise their responses as a gold standard against which proxies can be compared. The way in which staff interpret quality of life in this more communicative and able group may, however, be different to the way in which the interpret quality of life in advanced dementia.

Improving the quality of care for older people in long-term care has become a focus of attention for health and social care providers, both within the UK and internationally,[2] and an increasing number of evaluative research studies are being published that test the effectiveness and cost-effectiveness of interventions in this setting. Residents' quality of life (QoL) is frequently used as an outcome measure in these studies both to maintain a patient-centred focus and to facilitate health economic evaluation. The prevalent frailty and cognitive impairment in the care home population, however, means that collecting self-reported QoL measures from residents is challenging. As a response to this, proxy responses to QoL items have sometimes been used. For these a consultee, drawn from one of care home staff, or a relative or friend, answers questions on the resident's behalf. Using proxy respondents can be unreliable in care home settings. There may be lack of continuity of care home staff contact with individual residents due to shift working and staff turnover, and family and friends may not be well placed to judge QoL domains if they visit residents for only short periods.[6]

The EuroQoL questionnaire is a widely used preference-based health-related quality of life (HRQoL) measure suitable for use in economic evaluations. The EQ-5D-5L version measures HRQoL across five domains (mobility, self-care, usual activities, pain and anxiety/depression) with the scale for each domain ranging from level 1 (no problems) to level 5 (extreme problems). The responses from the five domains are converted to QoL index scores (utilities) generated from a given country's general population.[7] These index scores can be used to calculate quality-adjusted life years (QALYs), which are a measure of the person's state of health—one QALY equates to 1 year in perfect health. QALYs are calculated using the area under the curve[8] defined by utility scores at the different assessment points over the study period. The cost per QALY gained from an intervention when compared with usual care is the chosen cost-utility measure for determining eligibility for public support of the intervention through the UK National Health Service.[9]

The scale for the first version of EQ-5D had only three levels (EQ-5D-3L). EQ-5D-3L has been shown to have good construct validity for self-report[10] and has been used to measure QoL of older people living in their own homes and in care homes.[11] The 5L version was developed subsequently to deal with identified issues with sensitivity and a ceiling effect on the EQ-5D-3L which limited its ability to discriminate between health states, particularly in those with higher QoL.[12] There is also an EQ-5D visual analogue scale (VAS) used to assess overall health status, ranging from 0 (worst imaginable) to 100 (best imaginable). VAS is recognised to have specific strengths and weaknesses[13] but is recommended to be used routinely alongside the self-classification questionnaire by the EuroQoL group because of its usefulness in establishing global health status.[14]

It is recognised that the EQ-5D, in all its forms, is limited by consequence of being a generic measure that fails to take account of the difference in what constitutes 'QoL' in a long-term care setting. It doesn't take account of shifts in emphasis about what constitutes well-being as residents enter long-term care, which means that social care-related quality of life measures such as the Adult Social Care Outcomes Toolkit may be preferable in this setting.[15 16] A further critique has been that it fails to separate capability (what a resident is able to do) from preference (what a resident chooses to do under the circumstances), with the result that some authors have championed capability-based outcome measures, such as the ICEpop capability measure for older people, in care homes.[17 18] Best practice suggests that, if EQ-5D is used in this setting, it is used in combination with more specific instruments.

The R-outcome tool howRu has been specifically designed for use in long-term care settings in order to address QoL in a straightforward way that is practical with older people. HowRu is a patient-recorded outcome measure (PROM) that records four variables (pain or discomfort, feeling low or worried, limitation in activities and dependency on others) related to QoL at a fixed point in time ('How are you doing today?") on a four-point scale (none, slight, quite a lot, extreme).[19 20] The HowRu score is calculated by summing up the values for each domain to give a value on a 13-point scale ranging from 0 (worst) to 12 (best). The HowRu PROM was designed with older adults in mind[19 20] and may have a cogency and immediacy that improves on some of the measurement uncertainty introduced by the relative abstraction of the questions included in highly validated general population indices such as EQ-5D-5L.

In a comparison with EQ-5D in patients attending a cardiovascular outpatient clinic, HowRu was reported to have better readability, higher completion rate, wider range of states used and smaller ceiling effect.[17] No national tariffs exist to enable calculation of HowRu indices that would facilitate its use as a preference-based measure in cost-utility analysis. Understanding how and whether R outcomes correlate with EQ-5D scores in the care home setting is, however, helpful when considering additional information that can help to triangulate our understanding of how interventions affect QoL in this context. Knowing that HowRu correlates with EQ-5D may provide further justification for using it in clinical settings where broad judgements about impact on QoL have to be made without the need for detailed cost-utility analysis.

Proxies have been used to capture EQ-5D-3L responses from people with dementia, although poor agreement between patient and proxy ratings has raised concerns[15 16] and differences between professional and family carer ratings have led to questions about the appropriate choice of proxy.[16] In a comparison of clinicians and family carers as proxies, clinician responses had better construct validity in the more observable domains of mobility and self-care, and family carer responses had better construct

validity in the less observable domains of usual activities and anxiety/depression.[16] There is limited evidence, however, comparing self-reported and proxy responses to the EQ-5D-5L in care home populations.[17] There is, in particular, a paucity of data as to how it performs in UK care home populations. This is important because institutional care is structured differently between nations, with differing professional carer competencies, patterns of working and job roles. This means that carers in different countries will have differential exposure to residents and different competencies in terms of their ability to interpret residents' experiences, and a tool that works for professional proxy response in the USA may not, therefore, work as well in the UK.

The HowRu, as a recently developed PROM, is yet to be fully evaluated for older people living in care homes.[21] It is not known whether proxy responses in this setting may be useful in completing HowRU, and there are no data on how HowRu correlates with EQ-5D in the care home populations.

This study seeks to fill these evidence gaps.

## AIM

The Assessing Proxy Reliability In Care home Outcome Testing (APRICOT) study is a preparatory piece of work for the Proactive Healthcare for Older People Living in Care Homes (PEACH) study. It aims to examine the level of agreement between the responses to EQ-5D-5L and HowRu by care home staff and residents and between EQ-5D-5L and HowRu as QoL measures. Findings will assist in the interpretation of QoL data gathered for the larger PEACH study.

## OBJECTIVES

To determine the level of agreement between:
► Resident EQ-5D-5L and staff proxy EQ-5D-5L responses.
► Resident HowRu and staff proxy HowRu responses.
► Resident EQ-5D-5L and HowRu responses.
► Proxy EQ-5D-5L and HowRu responses.

## METHODS
### Setting

Twenty-four care homes in the East Midlands area of England. These are long-term care institutions, housing predominantly older people with frailty who can no longer be cared for at home. Detailed descriptions of the UK care home sector and the residents living within it have been published elsewhere.[4]

### Brief description of the PEACH study

The PEACH intervention involves using quality improvement collaboratives as a mechanism to encourage implementation of comprehensive geriatric assessment (CGA) as a unifying framework for assessment and delivery of healthcare in UK care homes. CGA is widely recognised as

a gold standard way to deliver care for older people with frailty.[22] The aim of PEACH is to improve quality of care and QoL for care home residents. Outcome data quantifying healthcare resource use and resident level QoL will be collected on a monthly basis to assess the impact of the intervention.

Two instruments are being used in PEACH to assess residents' QoL, the EQ-5D-5L and HowRu. The rationale is that these reflect measurable differences in the patient experience that may translate, with some interpretation, into an understanding of how CGA influences quality of care and general well-being. APRICOT has been designed as a preparatory substudy within PEACH to better enable interpretation of proxy EQ-5D-5L and HowRU responses collected as part of outcome measurement.

### Participants

Care home recruitment for PEACH took place between October 2016 and January 2017, with individual resident recruitment from January 2017. A prospective cohort of a subpopulation of residents will be included in the comparison of proxy and self-report measurement of EQ-5D-5L and HowRu in APRICOT. Residents included in the study will be those ≥60 years across 24 care homes and not receiving short stay or respite care. To have a better reflection of self-reported and proxy agreement in a care home setting, we will include residents with and without mental capacity. Care home managers will provide guidance with regards to residents with and without capacity to participate. When managers are unsure, researchers will make judgements based on the framework for mental capacity outlined in the 2005 Mental Capacity Act for England and Wales[23] and in keeping with the recommendations of that Act for inclusion research, for residents that lack capacity to provide consent to participation, an appropriate person will be consulted to make a decision with regards to participation in the study.

This study will be conducted in parallel to the main PEACH study. In addition to the routine collection of EQ-5D-5L and HowRu from residents recruited to PEACH, proxy responses to EQ-5D-5L and HowRu will be gathered from staff. We will include staff such as care home assistants, care home manager and registered nurses, who know the resident well. We will exclude staff employed in a supportive role, such as activity coordinators, since their orientation to supporting residents is more variable.

### Data collection

Data from proxies will be collected on three consecutive months. Due to the high staff turnover among care home staff, and to enable the influence of carer characteristics on the level of agreement to be estimated, data on the carer will be gathered at each assessment. Repeated measures are required for the final analysis in the PEACH study for calculating costs per QALY gained (comparing the intervention with usual care condition) and understanding

how the agreement changes at different time points is therefore of interest.

Staff proxies will be asked to consider the proxy-resident's perspective when completing the questionnaire using the following statement: 'Please rate how you (staff) think the resident will rate his/her own health-related quality of life, if the resident was to communicate.[24] Both self-reported and proxy EQ-5D will be completed on the same day to minimise any variations in responses.

The EQ-5D VAS will be used in the study in keeping with the recommendations of the EuroQoL group.

## Primary analysis

An overall agreement between the self-reported and staff proxy responses on the domain levels of the EQ-5D-5L and HowRu will be computed. Weighted kappa statistic and intraclass correlation (ICC) will be used to calculate the level of agreement for categorical and continuous outcomes respectively. All reliability indices will be calculated at the domain levels and overall index scores/QALYs for the EQ-5D-5L. To calculate the EQ-5D-5L index scores, responses from the descriptive system will be transformed into index scores derived from the UK general population. This will be done using the crosswalk value set,[25] which maps the 5L descriptive system data onto the 3L valuation.

Weighted kappa helps to distinguish between small and large difference in agreement ratings assigned to the different levels of each domain, but equal importance is given to disagreement.[26 27] The weighting for kappa will be done using linear weight—this assigns the same importance to the difference between any two categories within the response scale.[28] The CI for the weighted kappa will be calculated by bootstrapping in Stata 15 (Statcorp, 2015) with 1000 replications. This will be done for each time point.

The kappa statistic ranges from –1 to 1, and the strength of the agreement will be interpreted with regards to published guidelines[29] with agreement being:
► Poor, if kappa ≤0.00.
► Slight, if kappa=0.00 to 0.20.
► Fair, if kappa=0.21 to 0.40.
► Moderate, if kappa=0.41 to 0.60.
► Substantial, if kappa=0.61 to 0.80.
► Almost perfect, if kappa ≥0.80.

Unadjusted ICC will be calculated using two-way mixed effect Analysis of Variance (ANOVA) model to examine the level of agreement between the self-reported and proxy responses for the EQ-5D-VAS, EQ-5D index scores and total QALYs. ANOVA models are reported to be robust to deviations in normality and have been used in other QoL agreement studies.[16 30] The Bland-Altman graph (plotting the mean difference between the EQ-5D-5L-S (self-report) and EQ-5D-5L-P (proxy) against the mean of the two measures) will be constructed to supplement the ICC.

Analysis will be done at each time point for kappa and ICC. However, a single ICC value for QALYs will be calculated for individuals with data on all three consecutive months as this will be used in practice in the PEACH study, where analysis will be done on consecutive measures made over time.

To allow for comparability of the level of agreement at the domain and index score level; the same benchmarks used for kappa will be used for the ICC.

## Clustering

Clustering will be adjusted for because the calculation of kappa and ICC assumes independence of observations. In our study, clustering could occur at three levels at each time point. First, at the care home level, where residents within the same care home have similar characteristics and are different from those in other care homes. Second, at the individual level, where responses are clustered within each resident and lastly, at staff level, where staff members within a care home responds on behalf of multiple residents.

The ICC value will be calculated allowing for clustering using a nested two-way mixed effect model calculated by fitting a two-level random effect model with a random effect for care home and individuals.

A cluster-adjusted kappa will be calculated using a variance formula. This will include calculating kappa and its variance for each care home, then estimating the within cluster variance $\sigma_\omega^2$ using the average of the individual variances, and between cluster variance $\sigma_b^2$ using the variance of the individual kappa. Using these estimates, the clustered kappa and its variance will be calculated using the formulas below:

$$\omega_i = \frac{n_i}{(1+(n_i-1)\rho_\kappa)}$$

Where
$n_i$ =size of cluster $i$
$\rho$ =intracluster correlation coefficient for kappa

$$\rho_\kappa = \frac{\sigma_b^2}{(\sigma_\omega^2+\sigma_b^2)}$$

Variance of the cluster-adjusted kappa will be obtained using the equation below[24]:

$$\sigma_{clustered\ \kappa}^2 = \frac{\Sigma_{i=1}^\kappa\ \omega_i^2(\sigma_b^2+\frac{\sigma_\omega^2}{n_i})}{(\Sigma_{i=1}^\kappa\ \omega_i)^2}$$

The clustered kappa and its variance will then be divided by the square root of the number of individuals to get the SE. The 95% CI will be calculated using this SE.

Cluster-adjusted kappa (EQ-5D-5L and HowRu domain levels) and ICC (EQ-VAS and EQ-5D index scores) will be reported for each time point. However, QALYs will be presented over time, 3 months, similar to how it will be calculated in the PEACH study.

## Sample size calculation

We need a sample size of 160 residents assuming a kappa of 0.145 and a confidence level width of 0.153 derived

**Table 1** Planned pair-wise alignment of HowRU and EQ-5D-5L domains for agreement analysis using kappa

| HowRu domains | EQ-5D-5L domains |
| --- | --- |
| Pain or discomfort | Pain/discomfort |
| Feeling low or worried | Anxiety/depression |
| Limited in what you can do | Mobility<br>Self-care<br>Usual activities |
| Dependent on others | Mobility<br>Self-care<br>Usual activities |

from a previous study,[16] given that 50% of residents will have any problems.

## Secondary analyses

The effect of age, sex and length of stay in care home (for residents), length time working in care of older people/care homes and role/rank (for staff) at baseline on the difference between staff and proxy EQ-5D-5L-S scores will be analysed using a multilevel mixed effect regression model.

To investigate the reliability of using HowRu as a QoL measure in the care home population compared with EQ-5D-5L, we will assess agreement between these indices using weighted kappa statistics. This will involve testing the level of agreement between domains with similar construct on both scales[31] as shown in table 1.

## Patient and public involvement

The APRICOT and PEACH studies were developed and designed in discussion with both care home sector and patient and public involvement (PPI) representatives. The initial research proposal and protocol was presented, prior to submission for funding, to the Dementia and Frail Older Person's PPI group hosted in the Division of Rehabilitation and Ageing, University of Nottingham. Amendments were made to the proposal and protocol in light of their feedback. The PEACH study team includes one PPI and two care home sector representatives who are present at all study management meetings, with oversight for the APRICOT sub-study. We keep all participating care homes working with PEACH updated through quarterly newsletters which will include dissemination of our findings in lay terms as these become available.

## Ethics and dissemination

This study is part of preparatory work for the larger PEACH study. The PEACH study protocol has been reviewed as part of good governance by the Nottinghamshire Healthcare Foundation Trust. We aim to publish this study in a peer-reviewed journal with international readership and disseminate it further using relevant national stakeholder networks and specialist societies.

**Author affiliations**
¹Division of Medical Sciences and Graduate Entry Medicine, School of Medicine, University of Nottingham, Derby, UK
²Division of Epidemiology and Public Health, School of Medicine, University of Nottingham, Nottingham, UK
³East Midlands Academic Health Science Network, Nottingham, UK
⁴School of Economics, University of Surrey, Guildford, UK
⁵Institute of Mental Health, University of Nottingham, Nottingham, UK
⁶Division of Rehabilitation and Ageing, University of Nottingham, Nottingham, UK
⁷East Midlands Collaboration for Leadership in Applied Health Research and Care, Nottingham, UK
⁸National Institute for Health Research (NIHR) Nottingham Biomedical Research Centre, Nottingham
⁹School of Health Sciences, City, University of London, London, UK

**Acknowledgements** The authors would like to acknowledge the broader PEACH study team: Mr Zimran Alam, Ms Anita Astle, Professor Tony Avery, Dr Jaydip Banerjee, Professor Clive Bowman, Dr Neil H Chadborn, Mr Michael Crossley, Dr Reena Devi, Professor Claire Goodman, Professor Pip Logan, Professor Finbarr Martin, Professor Julienne Meyer, Dr Dominick Shaw, Professor David Stott and Dr Maria Zubair.

**Contributors** All authors meet the ICJME criteria for authorship. AU, SL, KH-S, AL, GH, JJ, HG, TD, JRFG, ALG: conceived the study at a PEACH study management meeting and specified the aims and objectives. AU, SL, ALG: produced the initial draft of the protocol. KH-S, AL, GH, JJ, HG, TD, JRFG: contributing to subsequent redrafts. AU, SL: led on aspects of statistical design. JJ, HG: provided specialist health economics input. AU, SL, KH-S, AL, GH, JJ, HG, TD, JRFG, ALG: all reviewed the final manuscript and approved it prior to submission.

**Funding** This work has been conducted as part of the Proactive Healthcare of Older People in Care Homes (PEACH) study, supported by the Dunhill Medical Trust, award number FOP1/0115.

**Competing interests** None declared.

**Patient consent** Obtained.

**Ethics approval** Health Regulatory Authority and University of Nottingham Ethics Committee Advised that the Project Proceed as Service Development/Quality Improvement (correspondence available on request).

**Provenance and peer review** Not commissioned; externally peer reviewed.

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
