## [Reviewer comments · BMJ Open]

ARTICLE DETAILS

TITLE (PROVISIONAL)	Measuring health related quality of life of care home residents, comparison of self-report with staff proxy responses for EQ-5D-5L and HowRu: Protocol for Assessing Proxy Reliability In Care home Outcome Testing
AUTHORS	Usman, Adeela; Lewis, Sarah; Hinsliff-Smith, Kathryn; Long, Annabelle; Housley, Gemma; Jordan, Jake; Gage, Heather; Denning, Tom; Gladman, John; Gordon, Adam

VERSION 1 – REVIEW

REVIEWER	Luciana Scalone CHARTA Foundation, Milan and Centre for Research on Public Health, University of Milano Bicocca, Monza, Italy
REVIEW RETURNED	09-Mar-2018

GENERAL COMMENTS	Dear authors, the protocol entitled “Measuring health related quality of life of care home residents, comparison of self-report with staff proxy responses for EQ-5D-5L and HowRu: Protocol for the APRICOT (Assessing Proxy Reliability In Care home Outcome Testing) Study” shows an interesting study about testing the properties of some instruments to assess HRQoL in elderly at home care. Although the aim and the methods of this project as reported are overall interesting, some revisions and possible adjustments on the study design could be made to improve its quality and final value, and also the manuscript requires a number of revisions to make clear all the relevant parts of the authors' work. I will report below main comments that in my opinion should be addressed. Abstract It should include more information about the rationale, objective and methods of the study, instead of a long paragraph about Ethics and Dissemination, which could be shorter and yet comprehensive. Introduction 1- Page 5, Lines 29-50: the description of the EQ-5D is not correct or not clear in some parts and also some references are not correct or require others focusing more on the topic of reference: 7, 8, 10 and 11 do not seem the best references for your statements, ref. 9 could be integrated with the one on NICE, and 12 is not sufficient to conclude that the VAS does not work well. Furthermore, I suggest you add other references supporting what you are saying and even better, to look for other sources showing differently its properties. 2- The 5L version of the EQ-5D was not developed to increase necessarily responsiveness (defined as capability of assessing changes where these are present), but to increase capability to
---

detect differences between different health states (e.g. a state described as 23321 with 5L shows a worse state than if assessed with 12311 with 3L) and reduce ceiling effect (e.g. the n. of states described as 11111 with 3L decrease when described with 5L). In any case, please add references for your statement about why the 5L version was produced.

3- Actually, the VAS has shown to work well in different studies and some times it could be event better that the descriptive system – for instance, research shows that this very simple index probably says more than the descriptive system, in which specific questions (domains) are introduced. It is possible that the VAS covers also other problems not specified in it and I think that this property of the VAS could be interesting above all in a study like this one.

Accordingly, I suggest you pay more attention with your analyses to understand the VAS capability to complete the EQ-5D descriptive system and/or the HowRu.

4- Page 5, lines 50-57: your statement about the reported weakness of the EQ-5D must be referenced. Also, if the EQ-5D fails to catch wellbeing, which is captureb by the HowRu, this must be clarified appropriately and with the right references. Actually, from the domains of the HowRu, which mostly cover the same construct of the EQ-5D, it does not seem that this instrument captures wellbeing more than the EQ-5D. Please clarify better this issue about the need for social care and about drawbacks and/or strengths of the instruments used in your project.

5- Page 6, lines 10-18: it seems here that the R – outcomes are two instruments, and then that one is the instrument and one is an index derived from the answer to the one instrument. Probably they are the same thing, but this is not clear to me. What is reported in line 52 suggests that the PROM is only one, but not clear yet. Also, the sentence “HowRU score is calculated etc” is not clear since the score = 13 depends on the original scores which are not clarified. Please rewrite and also provide a copy of the instrument(s) to be looked at.

6- Page 6, lines 18-19: are you sure that PROMS have been produced with older adults in minds? This could be true for some PROMS, but I doubt that this is generally true. Also, add a valid reference to this statement.

7- Page 6, lines 36-43: I think the references reported, 7, 8, 9 and 10, are wrong – you probably mismatched the numbers.

8- Page 6, lines 43-46: please add valid references.

Methods

9- Page 7, lines 35-38: the rationale reported refers to quality of care and wellbeing, however, it is not clarified how two HRQOL instruments can help understand these aspects, which can be related with but are not HRQOL. Is the HowRU a wellbeing instrument? If so, this is not clear in the paper.

10- Page 7, Line 40: is the APRICOT focusing only of proxy reported data?

11- Page 8, line 9: is the APRICOT focusing only of proxy reported data?

12- Page 8, line 14: I cannot understand the sentence “how the reliability of proxies change over subsequent measurements with therefore important”

13- Page 8, line 16: please add source and reference for the standard operating procedures you mention.

14- Page 8, lines 18-23: the statement specified to proxy responders in order to guide them when completing the questionnaires is probably not the one advised for the EQ-5D: please check and add

	the correct reference 15- Page 8, lines 23-25: since the order of administering the questionnaire may influence the responses, please take attention at reversing it in half of the study sample to control for this possible cause of bias. 16- Page 8, lines 16-27: please reconsider this statement after investigating well the literature about the performance of the VAS. Sample size calculation 17- This paragraph is not clear. Where are the number 0.153 and 0.145 derived from? Why do you use the weighted kappa as a parameter for calculating the sample size? Also, since this paragraph is before the section explaining what kind of analyses your plan to conduct, the reader has a higher difficulty to understand it. Please rewrite to make it clear. Statistical analyses 18- Please verify with a statistician who has experience in analyzing QoL data (hence not a statistician with no experience in this field) to understand if all the decisions made on the type of estimates you plan to use are correct and correctly applied on the data. For instance, I wonder whether the ICC is appropriate for data that are not normally distributed, as you must expect at least from the utility index. 19- You do not specify how the descriptive system answers will be converted into utility index, including what value set you plan to use and appropriate reference. Ethics and dissemination No mention about the informed consent and ethical approval by the participants.
--	---

REVIEWER	Stephanie Harrison Flinders University, Australia
REVIEW RETURNED	13-Mar-2018

GENERAL COMMENTS	This is an interesting protocol for a study to examine staff-proxy and resident responses to quality of life measures. I have included some comments for consideration:  -Could participants without the 'mental capacity to consent to participation' be included via a proxy consent process(e.g. a close family member)? By including only those who can consent for themselves many people living in care homes will not be eligible as you say 75-80% live with dementia. -Later in the protocol it is explained that only those who can consent for themselves were included because they are the gold standard. However, in this population I think it should be considered to include both those who can and cannot consent (although they may not be the gold standard they are a better reflection of people residing in care homes) and further justification to exclude those who cannot self-consent is needed. -Will you adjust for facility-level variables as there may be a clustering effect from people in the same care home -how will the staff be chosen, will they have known the resident for a certain amount of time, have had direct conversations or interactions with the resident, how will consistency in choosing the members of staff be ensured?
---

	Minor comments Introduction line 27: QOL should be changed to QoL
REVIEWER	Neslihan LOK Selcuk University Faculty of Health Sciences, Konya/Turkey
REVIEW RETURNED	27-Mar-2018
GENERAL COMMENTS	The study is a comprehensive and valuable work. However, some important points are not clear. There should be no abbreviation in the title. There should be some more details about where the work will be done. There are some procedural deficiencies in the study. Who will be the sample? What are the inclusion and exclusion criteria? How to make an application? What are the details of the application? A route map should be given to the application (eg a schema) Guidance for future studies seems a bit difficult as it is.

VERSION 1 – AUTHOR RESPONSE

Authors' responses.

Responses to Reviewer 1's comments

Thank you very much for your insightful comments. We have carefully revised our manuscript following your suggestions and we believe it is significantly enhanced as a consequence. Please see below our responses to comments one by one:

1. Abstract

It should include more information about the rationale, objective and methods of the study, instead of a long paragraph about Ethics and Dissemination, which could be shorter and yet comprehensive.

Response: Thank you. We have shortened the section on ethics and dissemination. We are otherwise satisfied with our abstract.

2. Introduction

The description of the EQ-5D is not correct or not clear in some parts and also some references are not correct or require others focusing more on the topic of reference: 7, 8, 10 and 11 do not seem the best references for your statements, ref. 9 could be integrated with the one on NICE, and 12 is not sufficient to conclude that the VAS does not work well. Furthermore, I suggest you add other references supporting what you are saying and even better, to look for other sources showing differently its properties.

Response:

In some ways this feedback was not entirely helpful. It would have been much more useful given the reviewer's intimate association with the EQ-5D measurement for her to have specified her preferred references. We have, however, reviewed our currently cited references one-by-one and responded as follows:

- We believe that reference 7 is appropriate. Our statement is that “QoL index scores (utilities) generated from a given country’s general population” – the cited paper describes how tariffs were developed based upon the UK’s general population. We are not sure which reference could be more appropriate.
- We accept that 8 was not the best reference regarding calculation of QALYs- we have replaced it with the following: Whitehead SJ, Ali S. Health outcomes in economic evaluation: the QALY and utilities. Br Med Bull. 2010;96(1):5–21.
- Reference 9 is “the one on NICE” – we are not sure what the peer reviewer meant by this comment.
- We accept that reference 10 was not well chosen, as it was a reference text. We found it difficult to decide which of the series of reports looking at construct validity of EQ-5D-3L in different populations and health conditions to cite. Eventually we opted for an article which reviews many of these papers as a compromise: Janssen MF, Simon Pickard A, Golicki D, Gudex C, Niewada M et al. Measurement properties of the EQ-5D-5L compared to the EQ-5D-3L across eight patient groups: a multi-country study. Qual Life Res. 2013; 22(7): 1717-27
- We believe that reference 11 fulfils what is required of it, which is to provide evidence of EQ-5D being used in older populations in community and care home settings. We would, however, be happy to cite an alternative illustrative example if the peer reviewer wishes to direct us to one.

3. Introduction

The 5L version of the EQ-5D was not developed to increase necessarily responsiveness (defined as capability of assessing changes where these are present), but to increase capability to detect differences between different health states (e.g. a state described as 23321 with 5L shows a worse state than if assessed with 12311 with 3L) and reduce ceiling effect (e.g. the n. of states described as 11111 with 3L decrease when described with 5L). In any case, please add references for your statement about why the 5L version was produced.

Response: We accept that the word “responsiveness” can be interpreted in the way that the peer-reviewer has done and we do not seek to mislead. We have therefore rewritten this section as follows: “The 5L version was developed subsequently to deal with identified issues with sensitivity and a ceiling effect on the EQ-5D-3L which limited its ability to discriminate between health states, particularly in those with higher quality of life” (Page 3, line 133). We have in addition cited the following useful review paper as a source, since it concisely reflects the numerous discussions in the literature that led up to the development of the 5L version: Devlin NJ, Brooks R. EQ-5D and the EuroQol Group: Past, Present and Future. Appl Health Econ Health Policy. 2017 Apr;15(2):127–37.

4. Introduction

Actually, the VAS has shown to work well in different studies and sometimes it could be even better than the descriptive system – for instance, research shows that this very simple index probably says more than the descriptive system, in which specific questions (domains) are introduced. It is possible that the VAS covers also other problems not specified in it and I think that this property of the VAS could be interesting above all in a study like this one. Accordingly, I suggest you pay more attention with your analyses to understand the VAS capability to complete the EQ-5D descriptive system and/or the HowRu.

Response: We accept this critique. We have removed our relatively narrow and potentially misleading critique of the VAS and have instead highlighted its usefulness in measuring global health status as part of the EQ-5D. We have modified our wording as follows: “VAS is recognised to have specific strengths and weaknesses(13) but is recommended to be used

routinely alongside the self-classification questionnaire by the EuroQoL group because of its usefulness in establishing global health status(14).” (page 3, line 137)

We have modified line 311 on page 6 to read as follows: “The EQ-5D VAS will be used in the study in keeping with the recommendations of the EuroQoL group.”

We have avoided further detailed critique of the VAS as this is primarily a protocol paper, rather than a detailed review of EQ-5D.

5. Introduction

Page 5, lines 50-57: your statement about the reported weakness of the EQ-5D must be referenced. Also, if the EQ-5D fails to catch wellbeing, which is captured by the HowRu, this must be clarified appropriately and with the right references. Actually, from the domains of the HowRu, which mostly cover the same construct of the EQ-5D, it does not seem that this instrument captures wellbeing more than the EQ-5D. Please clarify better this issue about the need for social care and about drawbacks and/or strengths of the instruments used in your project.

Response: We are afraid that the line and page reference cited by the reviewer did not match those on our document and so we have had to guess which statement about the weakness of EQ-5D needs referencing. We believe that the reviewer was referring to this text:

“It is recognised that the EQ-5D, in all its forms, is limited by consequence of being a generic measure that fails to take account of the difference in what constitutes “quality of life” in a long-term care setting. It doesn’t take account of shifts in emphasis about what constitutes wellbeing as residents enter long-term care, which means that social care related quality of life (SCRQoL) measures such as the Adult Social Care Outcomes Toolkit (ASCOT) may be preferable in this setting. A further critique has been that it fails to separate capability (what a resident is able to do) from preference (what a resident chooses to do under the circumstances), with the result that some authors have championed capability-based outcome measures, such as the ICEpop Capability Measure for Older People (ICECAP-O), in care homes (15-18).”

This is, as can be seen by the above text, already referenced (references 15-18). We have nothing more to add here.

6. Introduction

It seems here that the R – outcomes are two instruments, and then that one is the instrument and one is an index derived from the answer to the one instrument. Probably they are the same thing, but this is not clear to me. What is reported in line 52 suggests that the PROM is only one, but not clear yet. Also, the sentence “HowRU score is calculated etc” is not clear since the score = 13 depends on the original scores which are not clarified. Please rewrite and also provide a copy of the instrument(s) to be looked at.

Response: We have clarified our statements on the HowRu and HowRwe. Only the HowRu tool is used in APRICOT. HowRwe is a separate index and is not considered here. To avoid confusion, we have removed reference to it and modified the wording as follows:

“The R-outcome tool howRu has been specifically designed for use in long term care settings in order to address quality of life in a straightforward way that is practical with older people.” (page 4; line 163)

The tools are not our intellectual property and we do not have permission to share them in publication but the references cited direct the reader to the relevant source material, where the developers of the tool are cited as corresponding authors.

7. Are you sure that PROMS have been produced with older adults in minds? This could be true for some PROMS, but I doubt that this is generally true. Also, add a valid reference to this statement.

Response: We recognise that our syntax here was misleading. We were referring specifically to the HowRu prom, rather than PROMS more generally. We have modified our wording accordingly: “The HowRu PROM was designed with older adults in mind, and may have a cogency and immediacy that improves upon some of the measurement uncertainty introduced by the relative abstraction of the questions included in highly validated general population indices such as EQ-5D-5L.” (Page 4, line 170)

8. Introduction
I think the references reported, 7, 8, 9 and 10, are wrong – you probably mismatched the numbers.

Response: We think this refers to the citation of these references on page 4. These were in fact, misaligned when we removed the endnote coding from our document for submission. We have corrected these references now.

9. Methods
The rationale reported refers to quality of care and wellbeing, however, it is not clarified how two HRQOL instruments can help understand these aspects, which can be related with but are not HRQOL. Is the HowRu a wellbeing instrument? If so, this is not clear in the paper.

Response: This refers to the point, already addressed above, whereby we had described the HowRwe tool (which is a quality of care tool) alongside the HowRu PROM. We have, we think, resolved the confusion introduced by removing the reference to the HowRwe tool, which is not part of the APRICOT study. We have in the process, removed our mention of “quality of care” (page 4, line 163)

10. Is the APRICOT focusing only of proxy reported data?

Response: We would like to clarify that APRICOT focuses on both self-reported and staff proxy data as reported in its objectives (see page 6, line 239)

11. I cannot understand the sentence “how the reliability of proxies change over subsequent measurements with therefore important”

Response: Thanks a lot. We re-wrote this sentence to make it clearer and more understandable (page 6, line 302)

12. The statement specified to proxy responders in order to guide them when completing the questionnaires is probably not the one advised for the EQ-5D: please check and add the correct reference

Response: Many thanks for pointing this out. We have checked the statement for proxy responders as advised. As a result we re-wrote this and added the correct reference (Selai et al) (page 6, line 307)

13. Please add source and reference for the standard operating procedures you mention.

Response: Thank you for your comment. We acknowledge that this statement is confusing and have taken it out. By standard operating procedures we meant the statement specified to proxy responders to guide them in completing the questionnaire, which has been questioned in the comment above and addressed in our revision.

14. Please reconsider this statement after investigating well the literature about the performance of the VAS.

Response: Please see our above comments about how we have removed our narrow critique of the VAS and taken a more straightforward approach to its inclusion

15. Sample size calculation

This paragraph is not clear. Where are the number 0.153 and 0.145 derived from? Why do you use the weighted kappa as a parameter for calculating the sample size? Also, since this paragraph is before the section explaining what kind of analyses your plan to conduct, the reader has a higher difficulty to understand it. Please rewrite to make it clear.

Response: Thanks for your valuable suggestion. We have re-written this section to make it clear and have moved this paragraph after the analysis section to make it more understandable (see page 8 line 597)

16. Please verify with a statistician who has experience in analysing QoL data (hence not a statistician with no experience in this field) to understand if all the decisions made on the type of estimates you plan to use are correct and correctly applied on the data. For instance, I wonder whether the ICC is appropriate for data that are not normally distributed, as you must expect at least from the utility index.

Response: Thank you for your comments. We acknowledged that the EQ-5D utility index follows a non-normal distribution (Parkin et al., 2014). However, given that it is a numerical variable, the ICC is still the most appropriate index for agreement analysis (Watson et al., 2010). Concerning the appropriate method of analysing this non-normal data we will use an ANOVA model which is reported to be robust to deviations in normality (Glass et al., 1972; Harwell et al., 1992) and have been used in other quality of life agreement studies (Devine et al., 2014). Based on your advice we have re-written this sentence and added related references in our revision (see page 7 line 517)

17. You do not specify how the descriptive system answers will be converted into utility index, including what value set you plan to use and appropriate reference.

Response: Thank you for this insightful comment. Responses from the descriptive system will be transformed into index scores derived from the UK general population using the crosswalk value set (Van Hout et al., 2012), which maps the 5L descriptive system data onto the 3L valuation. We have also added this comment to the analysis section (see page 7, line 493).

18. No mention about the informed consent and ethical approval by the participants.

Response: Ethical approval is already discussed. We have added a sentence on informed consent (page 10, line 658)

Responses to reviewer 2's comments:

We deeply appreciate your efforts in reviewing our manuscript. Thank you for your constructive comments and we have revised our manuscript accordingly. Responses to specific comments are given below:

Major comments:

1. Could participants without the 'mental capacity to consent to participation' be included via a proxy consent process (e.g. a close family member)? By including only those who can consent for themselves many people living in care homes will not be eligible as you say 75-80% live with dementia.

Response:

Thank you for your invaluable suggestion. Proxy consent is not the correct terminology as outlined in the research-related clauses of the UK Mental Capacity Act but we have highlighted that we will include residents without capacity to consent to participation via a close family member. We rewrote our inclusion criteria accordingly (page 6; line 284)

2. Later in the protocol it is explained that only those who can consent for themselves were included because they are the gold standard. However, in this population I think it should be considered to include both those who can and cannot consent (although they may not be the gold standard they are a better reflection of people residing in care homes) and further justification to exclude those who cannot self-consent is needed.

Response: Thank you for this insightful comment again. We acknowledged that the population should be a better reflection of care home residents rather than the gold standard. Therefore we will include both residents with capacity and without capacity to consent in the study. We have added this comment to the methods section (page 6; line 284).

3. Will you adjust for facility-level variables as there may be a clustering effect from people in the same care home -how will the staff be chosen, will they have known the resident for a certain amount of time, have had direct conversations or interactions with the resident, how will consistency in choosing the members of staff be ensured?

Response: Many thanks for pointing this out. Concerning your comments about adjusting for clustering effect we will be using a nested two-way ANOVA which will adjust for clustering at care home as mentioned in the analysis section (page 7; line 515). We acknowledged that we need to state our selection criteria explicitly, we have addressed this in our participants section of the methods (page 5; line 274 onwards).

Minor comment:

4. Introduction line 27: QOL should be changed to QoL

Response: Thanks for your attention. We have corrected this.

Responses to reviewer 3's comments:

We deeply appreciate your efforts in reviewing our manuscript. Thank you for your constructive comments and we have revised our manuscript accordingly. Responses to specific comments are given below:

1. There should be no abbreviation in the title.

Response: Thank you for your kind reminder. This has been corrected.

2. There should be some more details about where the work will be done.

Response: We have added the following sentence on line 246, page 5.

“These are long-term care institutions, housing predominantly older people with frailty who can no longer be cared for at home. Detailed descriptions of the UK care home sector and the residents living within it have been published elsewhere (4)”

3. What are the inclusion and exclusion criteria?

Response: Thank you for your insightful comment. We acknowledged that our inclusion and exclusion criteria with regards to residents and staff was not explicitly stated and thus unclear. We have stated this explicitly under the participant section of the methods (See page 5, line 274 onwards).

4. How to make an application?
5. What are the details of the application?
6. A route map should be given to the application (eg a schema) Guidance for future studies seems a bit difficult as it is.

Response: We are sorry, but we do not understand these questions and so are not able to respond to them.

Please note that in addition to these recommendations, we have added a section on PPI involvement on 9, in keeping with the updated requirements of the journal.

VERSION 2 – REVIEW

REVIEWER	Luciana Scalone University of Milano Bicocca, Italy
REVIEW RETURNED	29-May-2018

GENERAL COMMENTS	Dear Authors, Thank you for updating your paper considering my comments. I still have some comments for you, which I report below in CAPITAL LETTERS, in order to can consider and answer to them. 1. Introduction The description of the EQ-5D is not correct or not clear in some parts and also some references are not correct or require others focusing more on the topic of reference: 7, 8, 10 and 11 do not seem the best references for your statements, ref. 9 could be integrated with the one on NICE, and 12 is not sufficient to conclude that the VAS does not work well. Furthermore, I suggest you add other
--

references supporting what you are saying and even better, to look for other sources showing differently its properties.

Response:

In some ways this feedback was not entirely helpful. It would have been much more useful given the reviewer's intimate association with the EQ-5D measurement for her to have specified her preferred references. We have, however, reviewed our currently cited references one-by-one and responded as follows:

- We believe that reference 7 is appropriate. Our statement is that "QoL index scores (utilities) generated from a given country's general population" – the cited paper describes how tariffs were developed based upon the UK's general population. We are not sure which reference could be more appropriate.

REF 7 IS FROM 1995, AND IT SEEMS ATTACHED TO STATEMENT ABOUT THE EQ-5D-5L, WHICH IS MUCH YOUNGER. ADJUST THE TEXT – YOU SPEAK ABOUT THE EQ-5D INSTRUMENT AND ABOUT THE UTILITIES THAT CAN BE DERIVED FROM IT (REF 7) – THEN YOU CLARIFY THAT THE EQ-5D-5L WAS DEVELOPED MORE RECENTLY, WITH SPECIFIC REF.

- We accept that 8 was not the best reference regarding calculation of QALYs- we have replaced it with the following: Whitehead SJ, Ali S. Health outcomes in economic evaluation: the QALY and utilities. Br Med Bull. 2010;96(1):5–21. THIS REF SEEMS IDENTICAL TO THE ONE IN THE PREVIOUS VERSION OF THE PAPER

- We accept that reference 10 was not well chosen, as it was a reference text. We found it difficult to decide which of the series of reports looking at construct validity of EQ-5D-3L in different populations and health conditions to cite. Eventually we opted for an article which reviews many of these papers as a compromise:

Janssen MF, Simon Pickard A, Golicki D, Gudex C, Niewada M et al. Measurement properties of the EQ-5D-5L compared to the EQ-5D-3L across eight patient groups: a multi-country study. Qual Life Res. 2013; 22(7): 1717-27 THIS PAPER COMPARES PROPERTIES OF 5L VERSION AGAINST THE 3L VERSION, BUT YOU ARE SPEAKING ABOUT THE 3L VERSION. I SUGGEST YOU QUOTE SOME RIGHT PAPERS ON CONSTRUCT VALIDITY (PLEASE CHECK THAT THEY INCLUDE INFO ABOUT CONSTRUCT VALIDITY) OF THE 3L VERSION, BETTER MORE RECENT ONES, OR THOSE THAT COVER POPULATIONS THAT ARE SIMILAR TO YOUR TARGET POPULATION – IF YOU STATE THIS, YOU MUST TAKEN IT FROM OTHER LITERATURE OR NOT?

Page 5, lines 50-57: your statement about the reported weakness of the EQ-5D must be referenced. Also, if the EQ-5D fails to catch wellbeing, which is captured by the HowRu, this must be clarified appropriately and with the right references. Actually, from the domains of the HowRu, which mostly cover the same construct of the EQ-5D, it does not seem that this instrument captures wellbeing more than the EQ-5D. Please clarify better this issue about the need for social care and about drawbacks and/or strengths of the instruments used in your project.

Response: We are afraid that the line and page reference cited by the reviewer did not match those on our document and so we have had to guess which statement about the weakness of EQ-5D needs referencing. We believe that the reviewer was referring to this text:

	“It is recognised that the EQ-5D, in all its forms, is limited by consequence of being a generic measure that fails to take account of the difference in what constitutes “quality of life” in a long-term care setting. It doesn’t take account of shifts in emphasis about what constitutes wellbeing as residents enter long-term care, which means that social care related quality of life (SCRQoL) measures such as the Adult Social Care Outcomes Toolkit (ASCOT) may be preferable in this setting. A further critique has been that it fails to separate capability (what a resident is able to do) from preference (what a resident chooses to do under the circumstances), with the result that some authors have championed capability-based outcome measures, such as the ICEpop Capability Measure for Older People (ICECAP-O), in care homes (15-18).” This is, as can be seen by the above text, already referenced (references 15-18). We have nothing more to add here. I HAVE NOT READ THE REFERENCES 15-18 (PAST 13-15 IN YOUR PREVIOUS VERSION) BUT I BELIEVE THEY ARE CORRECT AS REGARDS THE STATEMENTS REPORTED. ACTUALLY, BY READING THE TEXT, THE FIRST STATEMENTS, SEPARATED BY FULL STOPS, DO NOT SEEM REFERENCED BY 15-18, WHICH APPEAR AT THE END OF THE THIRD STATEMENT. SO, YES, IT IS NOW CLEAR TO ME THAT YOU REFERENCED THEM, HOWEVER, PLEASE ADD THE APPROPRIATE REFERENCE(S) CLOSE TO THE RELATED SENTENCE. YOU CAN CONSIDER THAT THIS MISUNDERSTANDING COULD INVOLVE ALSO OTHER READERS. 2. Are you sure that PROMS have been produced with older adults in minds? This could be true for some PROMS, but I doubt that this is generally true. Also, add a valid reference to this statement. Response: We recognise that our syntax here was misleading. We were referring specifically to the HowRu prom, rather than PROMS more generally. We have modified our wording accordingly: “The HowRu PROM was designed with older adults in mind, and may have a cogency and immediacy that improves upon some of the measurement uncertainty introduced by the relative abstraction of the questions included in highly validated general population indices such as EQ-5D-5L.” (Page 4, line 170) ADD REFERENCE SHOWING YOUR STATEMENT ON “THE HOWRU PROM WAS DESIGNED WITH OLDER ADULTS IN MIND”, MAYBE THE SAME OF POINT ABOVE (6)?
--	--

REVIEWER	Stephanie Harrison Flinders University, Australia
REVIEW RETURNED	09-May-2018

GENERAL COMMENTS	Thank you for addressing my comments
--------------------------------------

VERSION 2 – AUTHOR RESPONSE

Thank you for the opportunity to respond to further comments from the reviewers. Referee 1 has been somewhat clearer in her critique, written in capitals, this time. This has made it easier to address her

concerns.

We have addressed these as follows:

1. REF 7 IS FROM 1995, AND IT SEEMS ATTACHED TO STATEMENT ABOUT THE EQ-5D-5L, WHICH IS MUCH YOUNGER. ADJUST THE TEXT – YOU SPEAK ABOUT THE EQ-5D INSTRUMENT AND ABOUT THE UTILITIES THAT CAN BE DERIVED FROM IT (REF 7) – THEN YOU CLARIFY THAT THE EQ-5D-5L WAS DEVELOPED MORE RECENTLY, WITH SPECIFIC REF.

Now that this point has been clarified we have found a more appropriate reference to replace reference 7 as follows: Devlin N, Shah K, Feng Y, Mulhern B, van Hout B. Valuing health-related quality of Life: An EQ-5D-5L Value Set for England. Health Economics. 2017;1-16 This is the up-to-date dataset for the UK EQ-5D-5L

2. THIS REF SEEMS IDENTICAL TO THE ONE IN THE PREVIOUS VERSION OF THE PAPER It's not. The first paper cited the reference "Matthews JN, Altman DG, Campbell MJ, Royston P. Analysis of serial measurements in medical research. BMJ. 1990;300(6719):230-235.", we replaced this with, "Whitehead SJ, Ali S. Health outcomes in economic evaluation: the QALY and utilities. Br Med Bull. 2010;96(1):5–21" The peer-reviewer is incorrect in their assertion.

3. Qual Life Res. 2013; 22(7): 1717-27 THIS PAPER COMPARES PROPERTIES OF 5L VERSION AGAINST THE 3L VERSION, BUT YOU ARE SPEAKING ABOUT THE 3L VERSION. I SUGGEST YOU QUOTE SOME RIGHT PAPERS ON CONSTRUCT VALIDITY (PLEASE CHECK THAT THEY INCLUDE INFO ABOUT CONSTRUCT VALIDITY) OF THE 3L VERSION, BETTER MORE RECENT ONES, OR THOSE THAT COVER POPULATIONS THAT ARE SIMILAR TO YOUR TARGET POPULATION – IF YOU STATE THIS, YOU MUST TAKEN IT FROM OTHER LITERATURE OR NOT?

We disagree. This paper is produced by the EQ-5D group and includes a very useful review of the work undertaken to date to validate both versions of the EQ-5D. It is more useful to the reader than one or more narrower paper considering construct validity in a single population. We remind the peer-reviewer that this is a protocol paper and the purpose here is to establish context in a defensible way, rather than to comprehensively review the history of the EQ-5D-3L or EQ-5D-5L.

3. I HAVE NOT READ THE REFERENCES 15-18 (PAST 13-15 IN YOUR PREVIOUS VERSION) BUT I BELIEVE THEY ARE CORRECT AS REGARDS THE STATEMENTS REPORTED. ACTUALLY, BY READING THE TEXT, THE FIRST STATEMENTS, SEPARATED BY FULL STOPS, DO NOT SEEM REFERENCED BY 15-18, WHICH APPEAR AT THE END OF THE THIRD STATEMENT. SO, YES, IT IS NOW CLEAR TO ME THAT YOU REFERENCED THEM, HOWEVER, PLEASE ADD THE APPROPRIATE REFERENCE(S) CLOSE TO THE RELATED SENTENCE. YOU CAN CONSIDER THAT THIS MISUNDERSTANDING COULD INVOLVE ALSO OTHER READERS.

We accept this critique. We have now split these references.

4. ADD REFERENCE SHOWING YOUR STATEMENT ON “THE HOWRU PROM WAS DESIGNED WITH OLDER ADULTS IN MIND”, MAYBE THE SAME OF POINT ABOVE (6)?

We have cited the only two published references on the HowRU. But we have recited them adjacent to this statement to mollify the peer-reviewer. It now appears, to our eye, somewhat clumsy to have

the same two references cited within a couple of lines of each other but we are happy to leave this to the discretion of the editorial and peer review team at BMJ Open.

VERSION 3 – REVIEW

REVIEWER	Luciana Scalone Research Centre on Public Health (CESP), University of Milano Bicocca, Via Cadore 48, I-20900 Monza, Italy
REVIEW RETURNED	18-Jul-2018
GENERAL COMMENTS	I thank the authors for their work and for answering to all my comments. I do not totally agree with their answer on some references used but see their valuable work and that the paper has reached a good level to be published. I congratulate for the results reached and hope to see this paper published.